# Spindle checkpoint activation by fungal orthologs of the *S. cerevisiae* Mps1 kinase

Amy Fabritius[1]*, Anabel Alonso[2], Andrew Wood[3], Shaheen Sulthana[1], Mark Winey[1]

**1** Department of Molecular and Cellular Biology, University of California, Davis, CA, United States of America, **2** BOND PET FOODS, Boulder, CO, United States of America, **3** Research and Development, Mawi DNA Technologies LLC, Pleasanton, CA, United States of America

* asfabritius@ucdavis.edu

**Data Availability Statement:** All relevant data are within the manuscript and its Supporting Information files.

**Funding:** This work was supported by UC Davis and P01 NIH (GM105537 to M.W.). The funders

## Abstract

There is an ongoing need for antifungal agents to treat humans. Identification of new antifungal agents can be based on screening compounds using whole cell assays. Screening compounds that target a particular molecule is possible in budding yeast wherein sophisticated strain engineering allows for controlled expression of endogenous or heterologous genes. We have considered the yeast Mps1 protein kinase as a reasonable target for antifungal agents because mutant or druggable forms of the protein, upon inactivation, cause rapid loss of cell viability. Furthermore, extensive analysis of the Mps1 in budding yeast has offered potential tactics for identifying inhibitors of its enzymatic activity. One such tactic is based on the finding that overexpression of Mps1 leads to cell cycle arrest via activation of the spindle assembly checkpoint. We have endeavored to adapt this assay to be based on the overexpression of Mps1 orthologs from pathogenic yeast in hopes of having a whole-cell assay system to test the activity of these orthologs. Mps1 orthologous genes from seven pathogenic yeast or other pathogenic fungal species were isolated and expressed in budding yeast. Two orthologs clearly produced phenotypes similar to those produced by the overexpression of budding yeast Mps1, indicating that this system for heterologous Mps1 expression has potential as a platform for identifying prospective antifungal agents.

## Introduction

Fungal infections are on the rise for a variety of reasons and pose a serious health risk, especially for the increasing population of immune compromised patients. Currently, the rise in *Candida auris* hospital infections is raising alarms (CDC News Release: https://www.cdc.gov/media/releases/2023/p0320-cauris.html). Furthermore, fungal infections may be contributing to some disease states, such as cancers, in ways that have not been previously appreciated [1]. However, the collection of antifungal agents is limited, and few new antifungals have been reported in recent years. Antifungal targets generally fall into two classes: fungal specific (cell wall and ergosterol) and conserved, with enough divergence that the host is protected (e.g. microtubules), with most falling within the first class. Most new antifungals treatments are combination therapies or next generation formulations of current antifungals that attempt to

had no role in the study design, data collection and analysis, decision to publish, or preparation of the manuscript.

**Competing interests:** The authors have declared that no competing interests exist.

alleviate side-effects, drug-drug interactions, or medication delivery issues [2]. However, some novel "first in class" compounds are being tested [3].

A challenge in development of new antifungals is the fact that fungi are eukaryotes such that host toxicity is a serious concern. However, the availability of genome sequences for numerous pathogenic fungi gives investigators the opportunity to test the endogenous forms of antifungal targets with the hope of tuning the agent for a target pathogen's protein while limiting its recognition of the human ortholog if there is one. Additionally, most current anti-fungals suppress growth, but are not lethal to the fungi, which is one likely factor leading to the increase in drug resistance [4].

The yeast Mps1 kinase may be a good antifungal target because when inactivated by muta-tion or as a druggable form, the loss of activity leads to rapid cell death. This phenotype of Mps1 mutant strains is known to arise from the fact that the kinase is required at multiple points during the cell cycle to promote the assembly of the mitotic spindle and it is required in the spindle assembly checkpoint that arrests cell cycle progression in response to defects in the mitotic spindle. On the assembly side, Mps1 is needed for SPB duplication and for kinetochore assembly. On the checkpoint side, Mps1 contributes to mitotic checkpoint complex (MCC) formation by regulating localization and catalyzation of the MCC assembly (along with the SAC scaffolding proteins, BUB1 and MAD1-MAD2), blocking the APC/C and anaphase onset until spindle microtubules are properly attached at the kinetochores (Reviewed in [5]). Mps1 inactivation leads to failure in these dual roles leading to catastrophic events wherein cells do not form functional bipolar spindles yet fail to arrest in mitosis in response to this defect such that the cells continue through a failed mitosis that generates inviable aneuploid cells. The rec-ognition that cells can be rapidly killed by inactivating Mps1 has led to it being considered a potential target for cancer chemotherapies in addition to our and others' [6] interest in adapt-ing it as an antifungal target.

In addition to showing that inactivation of Mps1 is lethal to yeast cells, we have also shown that overexpression of Mps1 inhibits cell growth. Analysis of the cells whose growth was inhib-ited by Mps1 overexpression revealed that the cells arrested in mitosis as large-budded cells with metaphase mitotic spindles [7]. This phenotype is the result of Mps1 overexpression con-stitutively activating the spindle checkpoint causing a mitotic arrest. As discussed above, Mps1 is known to act in the checkpoint. This "gain-of-function" phenotype for Mps1 offers the opportunity to test potential inhibitors in a simple growth assay wherein the inhibition of over-expressed Mps1 would restore growth to the cells. We have demonstrated the utility of this concept using a druggable allele of Mps1. Mps1-as1 (analog-sensitive) has an engineered ATP binding site that allows binding of an ATP analog that will not bind other protein kinases [8]. Overexpression of Mps1-as1 arrests cell growth as expected, and treatment of these cells with the inhibitor restores growth. Using the system, we have validated this positive-growth, whole cell assay for Mps1 activity.

In this study, we have asked whether the Mps1 overexpression assay could be recapitulated with the overexpression of Mps1 orthologs from other organisms. In total, we tested overex-pression of eight different fungal Mps1 genes, four of which are the Mps1 orthologs in human pathogens. We also tested the Mps1 orthologs from two yeasts that used in research, one of which is a plant pathogen, and two chytrid fungi. The Mps1 orthologs from two *Candida* strains, *C. auris* and *C. albicans*, were found to be toxic when overexpressed in budding yeast, but the others were not. This first analysis of heterologous Mps1 expression suggests that this approach has promise in that some Mps1 orthologs could recapitulate the cell cycle arrest seen with endogenous Mps1.

## Results

### Identification of Mps1 orthologs

Orthologs of the Mps1 protein kinase are found throughout most of the eukaryotic world (Fig 1). The general organization of these proteins is a long N-terminal extension that exhibits little conservation across phylogenies, a conserved protein kinase domain and a short C-terminal extension. The Mps1 protein kinase domain has some signature motifs and is in the general family of S/T/Y kinases [9]. For this study, we intended to focus on Mps1 protein kinase encoding genes from pathogenic yeast. The genomes of many such organisms are available and were searched as described in the materials and methods. The apparent Mps1 ortholog was identified in seven pathogenic yeasts or other fungi. The multiple alignment of the protein kinase domain in these genes is shown in S1 Fig.

DNA sequences encoding each of the Mps1 orthologs, codon-optimized for expression in *S. cerevisiae*, were synthesized and inserted into a plasmid, enabling expression in yeast from the inducible galactose promoter (see materials and methods). Expression of these genes from integrated constructs was confirmed by immunoblotting for the GFP tag and measuring GFP fluorescence intensity (S2 Fig). While GFP-Mps1 expression among different constructs and backgrounds was variable, the expression was consistently increased in the +Gal condition.

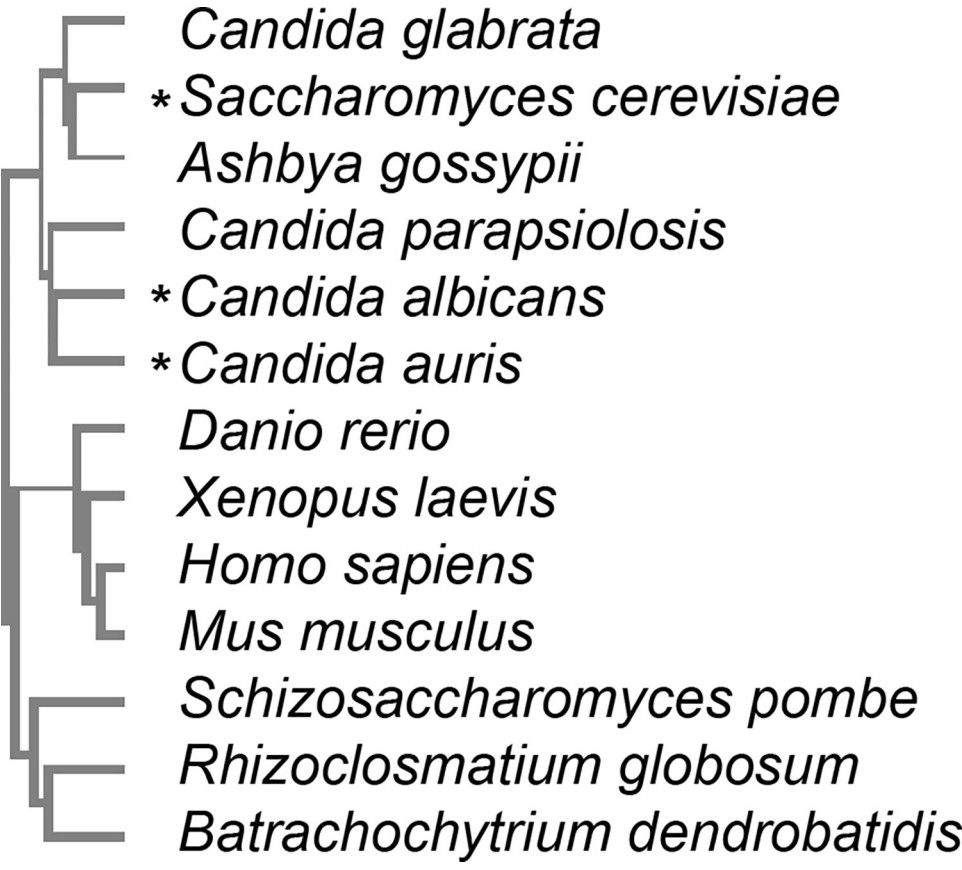

**Fig 1. Mps1 kinase phylogenetics.** Cladogram (guide) comparison of Mps1 kinase domains * indicates toxicity when overexpressed.

## Functional analysis of Mps1 orthologs in *S. cerevisiae*

As described above, a widely conserved function of Mps1 is its role in the spindle assembly checkpoint. The observation that Mps1 overexpression in *S. cerevisiae* led to a mitotic cell cycle arrest was an early indication of this function. The arrest is dependent on the spindle assembly checkpoint and arises from activation of the pathway which leads to high CDK kinase activity by inhibiting cyclin degradation by the APC. However, the arrest is transient because the cells are able to inhibit CDK (Cdc28) kinase activity via Swe1-mediated inhibitory phosphorylation. Overexpression of *S. cerevisiae* Mps1 in cdc28-VF strains, in which Cdc28 is insensitive to Swe1 inhibition is lethal [10] (Fig 2). We wished to determine whether any of the Mps1 orthologs were toxic in cdc28-VF strains upon overexpression. We found that, in addition to *S. cerevisiae* Mps1, overexpression of the Mps1 orthologs from either *Candida albicans*

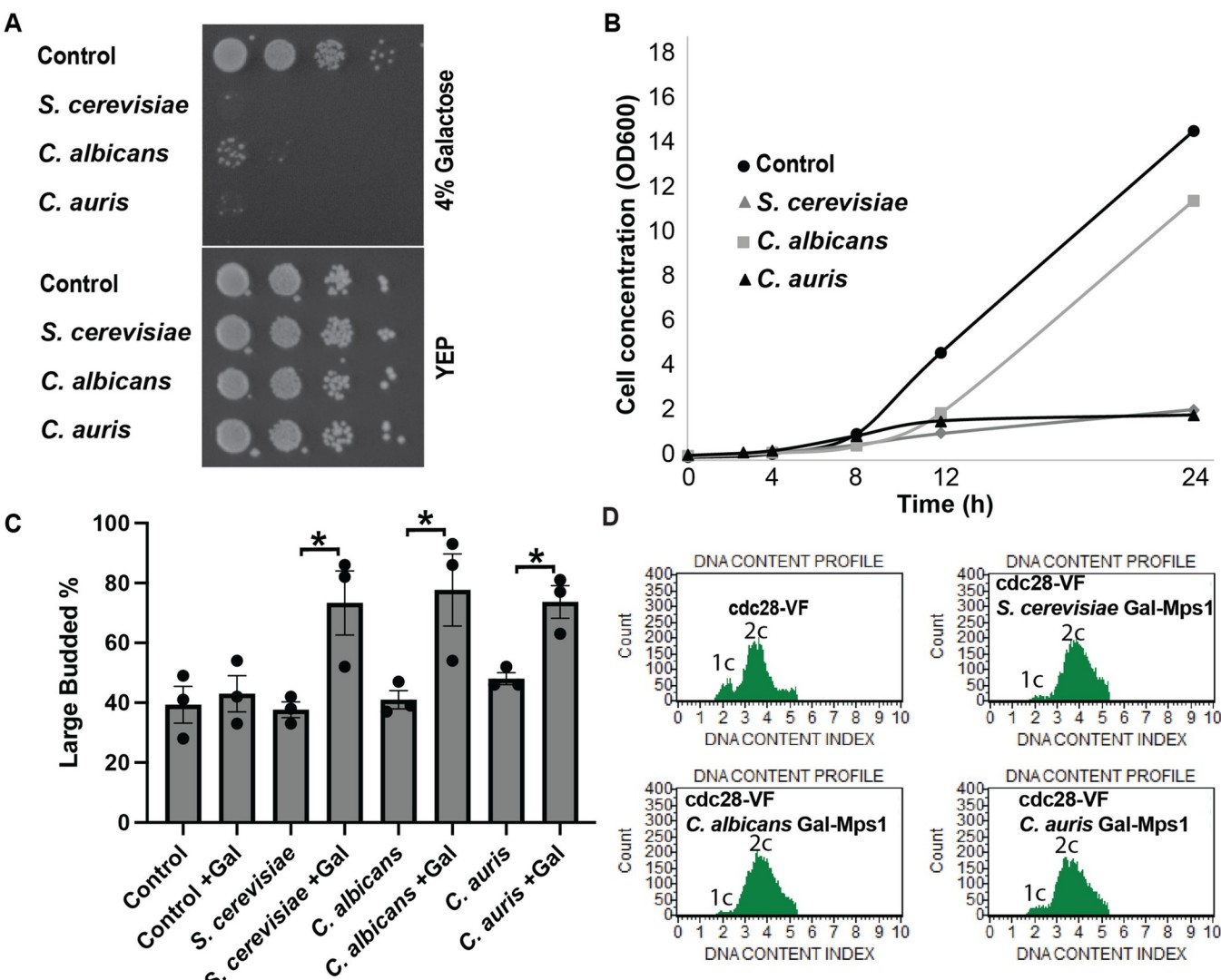

**Fig 2. Overexpression of Mps1 is toxic and causes a cell-cycle arrest in the cdc28-VF background.** All in cdc28-VF background, A. Colony growth on 4% galactose plates (Mps1 overexpression) and YEP plates (-galactose control)–dilution series. B. Growth curve of cells overexpressing Mps1 (4% galactose) of indicated species grown in liquid media. C. Percentage of large budded cells when overexpressing Mps1 of indicated species. D. Flow cytometric analysis of DNA content in cdc28-VF cells overexpressing indicated Mps1, showing cell-cycle arrest as single (2c) peak measured at 5 hours. *p<0.05.

or *Candida auris* was toxic in this *S. cerevisiae* strain in a plate assay (Fig 2A). The continued presence of some minimal growth (Fig 2A and 2B) in the *C. albicans* Mps1 strain may be due to cell variability and possibly loss of Mps1 overexpression in a subset of cells.

Inhibition of growth in a plate assay does not reveal the basis of the overexpressed gene's toxicity. We previously showed that *S. cerevisiae* Mps1 overexpression inhibited cell growth by causing a large budded mitotic cell cycle arrest. We asked whether overexpression of the *C. auris* or *C. albicans* Mps1 orthologs caused the same phenotypes. Fig 2 shows that overexpression of these Mps1 orthologs does inhibit cell growth based on optical density (Fig 2B) and that the cells arrest with large buds (Fig 2C). Finally, the spindle assembly mediated mitotic arrest is accompanied by cells containing G2/M post DNA synthesis quantities of DNA. Fig 2D shows the DNA content of cell populations as determined by flow cytometry. The cdc28-VF cells alone grow normally, but they do exhibit a bias toward mitotic G2/M DNA content. Nonetheless, a small population of G1 cells can be detected (Fig 2D, upper left panel). Overexpression *of S. cerevisiae* Mps1 in these cells eliminates this small peak, resulting in all of the cells having G2/M DNA content as expected for a mitotic arrest (Fig 2D upper right panel). This same result is observed in the cells overexpressing the *C. albicans* or *C. auris* Mps1 orthologs (Fig 2D, lower panels).

The complete panel of plate growth tests for all the strains expressing the various orthologs is shown in S3 Fig. We show growth with Mps1 overexpression in cdc28-VF strains, in addition to controls in wild-type cells and in a strain containing fluorescent protein markers for components of the microtubule cytoskeleton. The majority of the expressed orthologs cause no growth deficiency on plates. Furthermore, these orthologs did not cause a large-budded arrest (S3B Fig) or inhibit growth in liquid culture as assessed by optical density (S3C Fig).

Based on previous studies, we expected *S. cerevisiae* Mps1 to cause an arrest phenotype, but it appears that Mps1 from near relatives, but not of other fungal species, is able to trigger the arrest in this system. It is unclear whether the lack of phenotype by the overexpression of other species of Mps1 protein is due to their overexpression being insufficient or due to the *S. cerevisiae* host's inefficient recognition of the Mps1 substrates. However, because there was an arrest in some species, it suggests the system can work as a test for whether Mps1 could be a potential antifungal target for these species.

An Mps1 inhibitor has, in fact, been suggested as an antifungal for *C. albicans*. Use of the inhibitor decreased phosphorylation activity and reduced growth of fungal cells, while not affecting human Mps1 activity [6], an important factor in the design of antifungal treatments.

## Cellular structure impact of Mps1 ortholog overexpression

As discussed above, overexpression of *S. cerevisiae* Mps1 leads to mitotic arrest, but we have also reported that overexpression of a kinase-dead allele of Mps1 is toxic [8]. Furthermore, overexpression of the inactive Mps1 kinase has severe detrimental effects on spindle pole bodies and microtubule organization [11]. To assess the impact of Mps1 ortholog overexpression on cells, we overexpressed these constructs in strains containing fluorescent protein tagged microtubules (mCherry-Tub1) and spindle pole bodies (Nud1-mTurq). As expected, the overexpression of the Mps1 ortholog from *C. auris* or *C. albicans* caused a bi-polar, short spindle arrest as expected based on the above phenotypes that correspond to a mitotic arrest (Fig 3A and 3B). While Cdc28 is still presumably active in this background (mCherry-Tub1;Nud1-mTurq), the arrests may be due to a synthetic effect between the Mps1 overexpression and tagged cytoskeletal components.

Examination of mitotic spindles and spindle pole bodies in the strains not exhibiting toxicity or a cell cycle arrest upon overexpression of a Mps1 ortholog also did not reveal defects in spindle structures (e.g. Fig 3A).

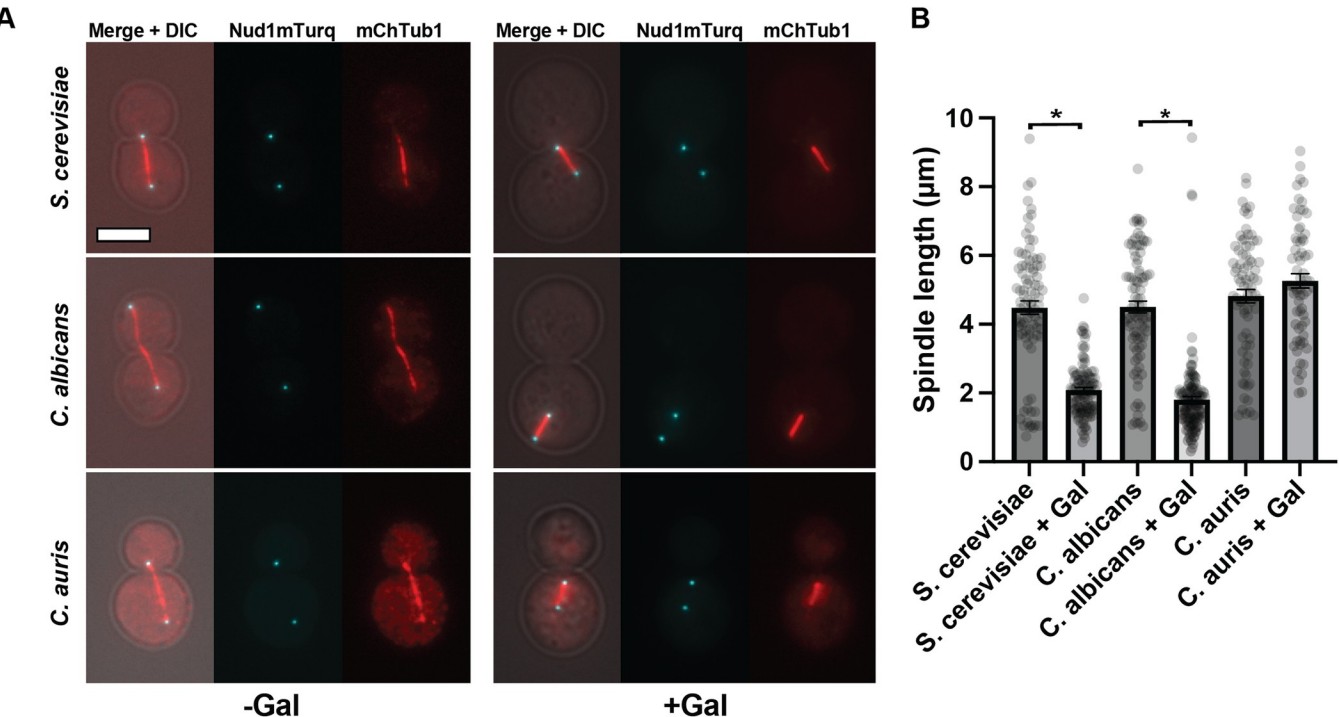

**Fig 3. Spindles in wild-type cells overexpressing Mps1 of indicated species.** A. Examples of long (left) and short (right) spindles in cells overexpressing Mps1 of indicated species in cells expressing mCherry-Tub1 (tubulin) and Nud1-mTurquoise (SPB marker). B. Average spindle lengths (Nud1 pole to pole length) of cells overexpressing Mps1 of indicated species. Scale bar = 4μm. *p<0.001.

## Conclusion

We report a cell-based assay for the function of Mps1 protein kinases encoded by orthologs from pathogenic yeasts. Conceptually similar to a "humanized mouse," we are using the innocuous *S. cerevisiae* to test gene function from pathogens. The assay is based on understanding the function of Mps1 in the spindle checkpoint of budding yeast. We have previously reported that Mps1 over-expression inhibits cell cycle progression by ectopically activating the spindle assembly checkpoint. This phenotype can be reproduced by over-expression of Mps1 from *Candida* species that are closely related the budding yeast, *Saccharomyces cerevisiae*. These *Candida* Mps1 orthologs include the gene from *C. auris*, a pathogenic yeast of growing concern as an infectious agent.

## Materials and methods

### Ortholog identification and comparison

Orthologs of *S. cerevisiae* Mps1 were identified by doing a BLASTp search targeted to the database for the organism of interest, and a reciprocal best hit protein BLAST search was used to confirm the identity. Kinase domains were determined based on amino acid sequence alignment to the known kinase domain of *S. cerevisiae* Mps1 [9]. Clustal omega was used to create a guide cladogram, with a real branch lengths comparison of kinase domain sequences.

### Gene synthesis and vector design

Double stranded DNA of codon optimized (idtdna.com/CodonOpt) mps1 of the indicated species was synthesized (IDT) and cloned into the XhoI and XbaI or XmaI sites of pSJ114 [12].

**Table 1. Strains and plasmid.**

| Strains | Genotype | Source or reference |
|---|---|---|
| yMW1844 | MATa cdc28-VF-HA | Adam Rudner |
| yMW5980 | MATa mCH-TUB1; NUD1-mTurquoise2-HIS3Mx ADE2 LEU2 | This study |
| yMW6073 | MATa LEU2+ mChTUB1; SPC42-mTurquoise2-SpHIS5; ura3::GAL-*MPS1-GFP-URA3 | This study |
| yMW6074 | MATa LEU2+ mChTUB1; SPC29-mTurquoise2-SpHIS5; ura3::GAL-*MPS1-GFP-URA3 | This study |
| yMW6006* | MATa ura3::GAL-*MPS1-GFP-URA3 | This study |
| yMW6013* | MATa cdc28-VF; ura3::GAL-*MPS1-GFP-URA3 | This study |
| yMW6026* | MATa mChTUB1; NUD1-mTurquoise2-HIS3Mx ADE2 LEU2; ura3::GAL-*MPS1-GFP-URA3 | This study |

*S. cerevisiae Mps1. Strains with Mps1 of other species only differ in Mps1 sequence.

## Yeast strains, transformation, culture

Yeast strains used in this study (Table 1) are derived from the w303 background. Plasmids were digested with ApaI and integrated into the *URA3* locus.

All yeast strains were grown at 30°C. Strains were grown to log phase in YEP containing 2% raffinose, then adjusted to OD600 of 0.1–0.4. For plate assays, tenfold serial dilutions were spotted onto glucose (repressing) or 4% galactose (expressing)(+Gal) plates. Plates were incubated at 30°C for 2–3 days.

For all other assays, galactose was added to 4% (Gal+) to induce GFP-Mps1 expression and cells were assayed after 5 hours of growth in galactose-containing medium.

## Cell cycle profiling

The cell cycle stage was determined using the Muse Cell Analyzer instrument and Muse Cell Cycle Kit (Millipore Sigma) according to the manufacturer's instructions.

For the budding index, cells were sonicated and were counted using a hemacytometer (N>50 cells per condition). Spindle length was measured as distance between Nud1mTurq foci (N>50 cells per condition). Statistical tests were done using a Student's t-test and error bars show the SEM.

## Fluorescence microscopy

Briefly, cells were fixed in 4% formaldehyde in 100 mM sucrose. Cells were sonicated, then images were acquired on an inverted microscope Eclipse Ti-E (Nikon) fitted with a CFI60 Plan Apochromat lambda 100x oil immersion objective lens (N.A. 1.45) (Nikon) and an ORCA-Flash4.0 V3 sCMOS camera. Data were acquired using NIS-Elements software. Image projections were made in FIJI [13,14].

Fluorescence intensity of GFP of a 0.65x0.65 μm (10x10 pixel) region between SPBs of short spindles was measured using FIJI (N = 10 cells per condition). Statistical tests were done using a Student's t-test and error bars show the SEM.

## Immunoblot analysis

Cells were collected by low-speed centrifugation, washed, and resuspended in B150 breaking buffer. Cells were then lysed by vortexing with glass beads for 10 min at 4°C. Extracts were clarified by centrifugation, and samples were run on an 8% SDS-PAGE gel. The membrane was blocked in 0.2% Tropix I-block reagent (Applied Biosystems) with 0.1% Tween-20 in PBS overnight at 4°C. The GFP tag was detected using a mouse monoclonal anti-GFP antibody (Biolegend) [14].

## Supporting information

**S1 Fig. Amino acid sequence alignment.** Amino acid sequence alignment of Mps1 kinase domains from indicated species.
(TIF)

**S2 Fig. Protein expression.** Mps1 Protein expression with (+) or without (-) galactose induction shown by A. Western blot in indicated cell background. Anti-GFP showing overexpression of indicted Gal-GFP-Mps1 construct. * = GFP-Mps1, arrows = internal control. B. GFP fluorescence intensity in nucleus measured from microscopy of cells expressing GFP-Mps1 in mCh-Tub1;Nud1-mTurq background **p<0.001.
(TIF)

**S3 Fig. Growth and viability with overexpression of Mps1.** A. Growth on dilution plates of cells overexpressing indicated species of Mps1 (4% galactose) or without overexpression (YEP) in three different cell backgrounds (cdc28-VF, WT, or mChTub1;Nud1mTurq). B. Budding indices with (+Gal) or without overexpression of indicated species of Mps1 in either a wild-type or mChTub1;Nud1mTurq background. C. Growth curve ($OD_{600}$) of cells overexpressing (4% galactose) indicated Mps1 in liquid culture in cdc28-VF background. *p<0.05.
(TIF)

**S1 Raw images.**
(TIF)

**S1 Raw data.**
(PDF)

## Author Contributions

**Conceptualization:** Anabel Alonso, Mark Winey.

**Data curation:** Amy Fabritius, Anabel Alonso, Andrew Wood, Shaheen Sulthana.

**Formal analysis:** Amy Fabritius, Anabel Alonso, Andrew Wood, Shaheen Sulthana.

**Investigation:** Amy Fabritius, Anabel Alonso, Andrew Wood, Shaheen Sulthana.

**Methodology:** Anabel Alonso, Mark Winey.

**Validation:** Amy Fabritius.

**Writing – original draft:** Amy Fabritius, Mark Winey.

**Writing – review & editing:** Amy Fabritius, Anabel Alonso, Andrew Wood, Shaheen Sulthana, Mark Winey.

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
