## [Decision Letter · Decision Letter 0]

8 Jan 2024

PONE-D-23-41815Spindle checkpoint activation by fungal orthologs of the S. cerevisiae Mps1 kinasePLOS ONE

Dear Dr. Fabritius,

Thank you for submitting your manuscript to PLOS ONE. After careful consideration, we feel that it has merit but does not fully meet PLOS ONE’s publication criteria as it currently stands. Therefore, we invite you to submit a revised version of the manuscript that addresses the points raised during the review process.

As you can see, the reviewers (one submitted the critique directly to me, not through the system; I append it below) took different views on the overall evaluation. However, they both found the work technically sound, and I believe the comments are easy to address with some editing, without new experiments. The work will be helpful in the field, and I look forward to receiving a revised version accommodating their suggestions. Because the edits are straightforward, I think a re-review will not be necessary.

We look forward to receiving your revised manuscript.

Kind regards,

Michael Polymenis, Ph.D.

Academic Editor

PLOS ONE

Journal Requirements:

3. Thank you for stating the following in the Acknowledgments Section of your manuscript: "This work was supported by the Supported by UCD and the P01 NIH (GM105537 to M.W.)".

Please remove any funding-related text from the manuscript and let us know how you would like to update your Funding Statement. Currently, your Funding Statement reads as follows: "The authors received no specific funding for this work."

4. We note that your Data Availability Statement is currently as follows: "All relevant data are within the manuscript and its Supporting Information files".

7. We notice that your supplementary figures are included in the manuscript file. Please remove them and upload them with the file type 'Supporting Information'. Please ensure that each Supporting Information file has a legend listed in the manuscript after the references list."

Additional Editor Comments:

As you can see, the reviewers (one submitted the critique directly to me, not through the system, I append it below) took different views on the overall evaluation. However, they both found the work technically sound, and the comments are easy to address with some editing, without new experiments. The work will be helpful in the field, and I look forward to receiving a revised version accommodating their suggestions.

(Additional review sent directly to the Academic Editor by another reviewer)

1. Recommendation: Major revision

2. Is the manuscript technically sound, and do the data support the conclusions?

- Yes, for the most part.

3. Has the statistical analysis been performed appropriately and rigorously?

- NA. Some scale bars need to be added to images.

4. Does the manuscript adhere to the PLOS Data Policy? Additional details can be found at http://journals.plos.org/plosone/s/materials-and-software-sharing.

- Yes

5. Is the manuscript presented in an intelligible fashion and written in standard English?

- Yes

6. Review Comments to the Author (minimum 200 characters)

- The manuscript is well written and logically presented. Much of the data is fine. However, there are several inconsistencies that need to be addressed and some of the supplementary data improved before publication.

Figure 1: Is part B necessary here? I am not convinced it is.

Figure 2

A: just a few colonies are growing in the strain expressing C.albicans. Is this reproducible? I wonder if these rare isolates are really expressing Mps1 still? They may account for most of the liquid growth in panel B (after 12 hours).

C. How does this VF dataset relate to S3 Fig.B, where apparently no large-budded arrests are seen in wild-type cells? The latter seems odd – wouldn’t mitotic delays be expected here too?

D: could the position of 1C and 2C control peaks (for the VF strain) simply be labelled here, on each plot? ‘DNA content index’ isn’t so helpful.

Figure 3. Scale bars should be added for these IF images.

Fig. S2

A: These blots are not ideal and in too many pieces. These should be improved or deleted.

The position of relevant molecular weight markers should be labelled.

B: The GFP quantitation, in panel B, more closely matches the western results from wild-type cells.

Was expression of the 5 other Mps1-GFP fusion proteins detected (by western), in the VF strains that did not show significant growth defects (Cg, Ag, Rg, Cp, Bd)?

Fig. S3

A: why does the wild-type strain over-expressing ScMps1 grow so well, when the Nud1/Tub1 equivalent does not?

B: why no large budded delay/arrest in GAL, in these WT strains? Cf Fig.2C and Nud1/Tub1 panel.

C: the gossypii-expressing strain delays/arrests as large budded (in the tagged Nud1/Tub1 strain) and colonies grow slowly. This doesn’t seem to happen in wild-type or VF. Could it be a synthetic phenotype with the tagged cytoskeletal markers (not fully functional)? Why is there no large-budded delay in the Nud1/Tub1 strain expressing C.auris?

B.d. overexpression looks unhappy, but only in the VF. Any large budded arrest there?

7. PLOS authors have the option to publish the peer review history of their article (https://journals.plos.org/plosone/s/editorial-and-peer-review-process#loc-peer-review-history). If published, this will include your full peer review and any attached files. Do you want your identity to be public for this peer review?

-Yes.

7a. If you answered "Yes" above, that you would like your identity to be public, please provide your full name, as it should appear on the published peer review. Please do not sign the review on behalf of another person.

-Kevin G. Hardwick

Reviewers' comments:

Reviewer's Responses to Questions

**Comments to the Author**

1. Is the manuscript technically sound, and do the data support the conclusions?

Reviewer #1: Partly

2. Has the statistical analysis been performed appropriately and rigorously? 

Reviewer #1: No

3. Have the authors made all data underlying the findings in their manuscript fully available?

Reviewer #1: Yes

4. Is the manuscript presented in an intelligible fashion and written in standard English?

Reviewer #1: Yes

5. Review Comments to the Author

Reviewer #1: The manuscript by Fabritius et al. describes the phenotype of overproduction of Mps1 fungal orthologs in budding yeast. The authors have found that heterologous expression of the MPS1 orthologs from C. albicans and C. auris with the GAL promoter leads to budding yeast cell lethality, the phenotype of which is comparable to that of overproduction of the endogenous Mps1. By observing the mitotic spindle morphology, the authors demonstrate that the cell growth defect caused by Mps1 and Mps1 ortholog overproduction is likely due to the activation of the mitotic spindle checkpoint. Surprisingly, overproduction of Mps1 orthologs from other related yeast species tested in this work show no detectable cell growth defect. Thus, the main conclusion of this work is limited, i.e. Mps1 from C. albicans and C. auris, which are closely related to budding yeast, function wise is comparable to budding yeast Mps1, whereas the other Mps1 orthologs tested by the authors are not. The experiments were conducted in a seemingly rigorous fashion, however, there is no experimental evidence presented in this work to support their main claim that heterologous expression of Mps1 orthologs in budding yeast can serve as a platform for drug screening.

The figure legends are very sketchy, they should be self-evident and detailed enough for the reader to understand the experiments.

Fig 2A should include a YEP panel as shown in Fig S3A.

Fig 3 needs quantification of spindle length and the number of cells observed.

Fig S2. Panel B is not the quantification of panel A, please explain the logic.

6. PLOS authors have the option to publish the peer review history of their article (what does this mean?). If published, this will include your full peer review and any attached files.

Reviewer #1: No

---

## [Author Response · Author response to Decision Letter 0]

6 Mar 2024

Thank you for your attention to our manuscript and for sharing the supportive and helpful feedback from the reviewers. Below we have listed the reviewer’s comments in italics followed by our answer to the comment or how we have addressed the issues preceded by “AU:.”

We have also submitted the requested copy of the revised manuscript with track changes indicated, and an unmarked version of the revised manuscript. 

Reviewer #1: The manuscript by Fabritius et al. ….. there is no experimental evidence presented in this work to support their main claim that heterologous expression of Mps1 orthologs in budding yeast can serve as a platform for drug screening. 

AU: We agree that there is no evidence directly supporting the claim of a drug screening platform. We present this as an idea of how to extend or apply these results. 

The figure legends are very sketchy, they should be self-evident and detailed enough for the reader to understand the experiments.

AU: The figure legends have been expanded to be more detailed.

Fig 2A should include a YEP panel as shown in Fig S3A.

AU: A YEP panel has been added to Fig 2A.

Fig 3 needs quantification of spindle length and the number of cells observed.

AU: Quantification of spindle length and number of cells observed has been added to Fig. 3 and to the methods section, respectively. 

Fig S2. Panel B is not the quantification of panel A, please explain the logic.

AU: Fig. S2 Panel B is not the quantification of A, panel B reports GFP signal. We report two different ways to document MPS1 expression, and this has been clarified in the revised figure legend. 

Reviewer #2/Kevin Hardwick:

Figure 1: Is part B necessary here? I am not convinced it is.

AU: Figure 1, part B has been removed, as it was determined to be unnecessary

Figure 2

A: just a few colonies are growing in the strain expressing C. albicans. Is this reproducible? I wonder if these rare isolates are really expressing Mps1 still? They may account for most of the liquid growth in panel B (after 12 hours).

AU: The few colonies growing in the plate assays of C. albicans is reproducible, and we can’t be certain that they’re still overexpressing MPS1. While possible that these “escapers” may be the cause of growth in the liquid growth assay (Fig2B) after 12 hours, it seems unlikely because of the short timeframe of 12 hours in liquid culture versus days on the plates. Nonetheless, this has been noted in the text. 

C. How does this VF dataset relate to S3 Fig.B, where apparently no large-budded arrests are seen in wild-type cells? The latter seems odd – wouldn’t mitotic delays be expected here too?

AU: While the VF strains (where CDC28 is insensitive to Swe1 inhibition) overexpressing MPS1 present a large-budded arrest (as well as reduced grown on plates and in liquid), this is not visible in a cycling population in a wild-type background expressing MPS1. In wild-type cells, fully functional Cdc28 allows the cells to bypass the arrest caused by increased MPS1 after a few hours. Our assays are performed with asynchronous cells growing 5 hours or more, allowing plenty of time for the cells to bypass the arrest with the aid of Cdc28. When Cdc28 is not regulated properly (i.e. cdc28-VF), the arrest persists, and we see the growth phenotypes, including large-budded arrests and decreased growth on plates and in liquid media. 

D: could the position of 1C and 2C control peaks (for the VF strain) simply be labelled here, on each plot? ‘DNA content index’ isn’t so helpful.

AU: The peaks in the cytometry plots have been labelled 1C and 2C for clarification.

Figure 3. Scale bars should be added for these IF images.

AU: Scale bars have been added to microscopy images.

Fig. S2

A: These blots are not ideal and in too many pieces. These should be improved or deleted.

The position of relevant molecular weight markers should be labelled.

AU: While the presentation of the Western blots is not ideal, we believe the information may be valuable to some readers and would like to share it as part of the supplemental information. Original full blots have been provided to the journal; the position of relevant molecular weight markers have been added.

B: The GFP quantitation, in panel B, more closely matches the western results from wild-type cells. Was expression of the 5 other Mps1-GFP fusion proteins detected (by western), in the VF strains that did not show significant growth defects (Cg, Ag, Rg, Cp, Bd)?

AU: GFP expression was sometimes variable among the strains, but we are hoping to show that the GFP-MPS1 expression is increased in the +Gal conditions. We have noted in the text that expression may be variable. Expression of the other 5 Mps1-GFP fusion was detected in some (but not all) VF strains that did not show significant growth defects. This was sometimes variable, and since there was not a clear growth phenotype, we did not further pursue establishing expression. 

Fig. S3

A: why does the wild-type strain over-expressing ScMps1 grow so well, when the Nud1/Tub1 equivalent does not?

AU: The better growth of the wild-type strain overexpressing ScMPS1 than the Nud1/Tub1 strain overexpressing ScMPS1 may be due to a synthetic phenotype caused by the addition of fluorescent markers to cytoskeletal components. 

B: why no large budded delay/arrest in GAL, in these WT strains? Cf Fig.2C and Nud1/Tub1 panel.

AU: Please see the above explanation concerning Fig 2C.

C: the gossypii-expressing strain delays/arrests as large budded (in the tagged Nud1/Tub1 strain) and colonies grow slowly. This doesn’t seem to happen in wild-type or VF. Could it be a synthetic phenotype with the tagged cytoskeletal markers (not fully functional)? Why is there no large-budded delay in the Nud1/Tub1 strain expressing C. auris?

AU: The gossypii-expressing Mps1 Nud1/Tub1 tagged strain being delayed/arresting as large- budded cells (but not in other backgrounds) may be due to a synthetic phenotype with the tagged cytoskeletal markers. It is unclear why C. auris Mps1 overexpression doesn’t cause a large-budded arrest in the Nud1/Tub1 background (while others do); but this lack of arrest is consistent with wild-type growth in the plate assay. It’s also surprising that there is an arrest in the other strains, as the plate growth phenotype would not suggest this result. However, this may be a short term arrest that is overcome in the longer-term plate growth assay – this arrest may be due to a synthetic phenotype with the Nud1/Tub1 tagging. 

B.d. overexpression looks unhappy, but only in the VF. Any large budded arrest there?

AU: The B.d. Mps1 overexpression in the cdc28-VF background was not checked for a large-budded arrest.

Please amend the Funding Statement as follows: 

This work was supported by UC Davis and P01 NIH (GM105537 to M.W.)

We appreciate the opportunity to submit a revised manuscript. We hope that we have been able to sufficiently address to the reviewer’s comments such that the manuscript is now acceptable for publication.

Please let us know if you have any further questions.

Sincerely,

Amy Fabritius, Ph.D., Project Scientist Mark Winey, Ph.D., Distinguished Professor

Molecular and Cellular Biology, UC Davis Molecular and Cellular Biology, UC Davis

---

## [Editor Report · Decision Letter 1]

11 Mar 2024

Spindle checkpoint activation by fungal orthologs of the S. cerevisiae Mps1 kinase

PONE-D-23-41815R1

Dear Dr. Fabritius,

We’re pleased to inform you that your manuscript has been judged scientifically suitable for publication and will be formally accepted for publication once it meets all outstanding technical requirements.

Kind regards,

Michael Polymenis, Ph.D.

Academic Editor

PLOS ONE
---

## [Editor Report · Acceptance letter]

18 Mar 2024

PONE-D-23-41815R1 

PLOS ONE

Dear Dr. Fabritius, 

I'm pleased to inform you that your manuscript has been deemed suitable for publication in PLOS ONE. Congratulations! Your manuscript is now being handed over to our production team.

Kind regards, 

on behalf of

Dr. Michael Polymenis 

Academic Editor

PLOS ONE